# Enhancing Road Safety: Fast and Accurate Noncontact Driver HRV Detection Based on Huber–Kalman and Autocorrelation Algorithms

**DOI:** 10.3390/biomimetics9080481

**Published:** 2024-08-09

**Authors:** Yunlong Luo, Yang Yang, Yanbo Ma, Runhe Huang, Alex Qi, Muxin Ma, Yihong Qi

**Affiliations:** 1School of Information Science and Technology, Southwest Jiaotong University, Chengdu 611756, China; yunlong.luo@my.swjtu.edu.cn; 2Pontosense Inc., Toronto, ON M5C3G8, Canada; 3Faculty of Computer and Information Sciences, Hosei University, Tokyo 184-8584, Japan

**Keywords:** millimeter-wave radar, heart rate variability, driver monitoring, signal processing, FMCW radar

## Abstract

Enhancing road safety by monitoring a driver’s physical condition is critical in both conventional and autonomous driving contexts. Our research focuses on a wireless intelligent sensor system that utilizes millimeter-wave (mmWave) radar to monitor heart rate variability (HRV) in drivers. By assessing HRV, the system can detect early signs of drowsiness and sudden medical emergencies, such as heart attacks, thereby preventing accidents. This is particularly vital for fully self-driving (FSD) systems, as it ensures control is not transferred to an impaired driver. The proposed system employs a 60 GHz frequency-modulated continuous wave (FMCW) radar placed behind the driver’s seat. This article mainly describes how advanced signal processing methods, including the Huber–Kalman filtering algorithm, are applied to mitigate the impact of respiration on heart rate detection. Additionally, the autocorrelation algorithm enables fast detection of vital signs. Intensive experiments demonstrate the system’s effectiveness in accurately monitoring HRV, highlighting its potential to enhance safety and reliability in both traditional and autonomous driving environments.

## 1. Introduction

### 1.1. Importance of Continuous Driver Health Monitoring

Monitoring a driver’s physical condition is crucial for enhancing road safety in both normal and autonomous driving scenarios. By continuously assessing the driver’s health, the system can prevent accidents caused by factors such as drowsiness and sudden medical emergencies like heart attacks. For fully self-driving (FSD) systems, this monitoring ensures that control remains with a capable driver, thereby maintaining safety [1,2].

This technology gains significance as automotive companies begin to commercialize FSD capabilities and consider offering insurance services. Ensuring the driver’s fitness to drive not only enhances safety but also reduces liability and insurance costs [3], providing a comprehensive safety net and financial benefit for both the company and consumers.

In addition to emergency health situations, driver drowsiness due to fatigue, alcohol consumption, and certain substances can lead to loss of vehicle control. Heart rate variability (HRV) serves as a critical indicator for detecting drowsiness and health-related emergencies [4,5,6]. By monitoring HRV, the system can identify early signs of driver impairment and take pre-emptive actions to prevent accidents, thus improving overall road safety and the reliability of FSD systems.

### 1.2. Technologies and Methods for HRV Monitoring

Technologies and methods for monitoring HRV play a pivotal role in ensuring effective driver health monitoring amidst the challenges of dynamic driving environments. Radio Frequency (RF) technologies, such as Wi-Fi-based channel status indicator (CSI) and radar-based millimeter-wave (mmWave) sensors, are employed for accurate HRV measurement [7,8]. The mmWave radar, operating at 60 GHz with frequency-modulated continuous wave (FMCW) technology behind the driver’s seat, has emerged as a preferred sensor for noncontact vital sign monitoring [9,10,11].

Continuous monitoring of a driver’s health during driving faces numerous challenges. These challenges include HRV displacement signals being mixed with respiration signals [12], which necessitates innovative approaches to separate the HRV signal effectively [13,14,15,16,17,18]. Dynamic driving conditions, such as road bumps and driver maneuvers, also pose challenges by causing unpredictable body movements relative to the sensor [13,19,20,21,22].

Quasi-static heart rate (HR) monitoring, conducted when the automobile is in its parking or starting phases, can significantly reduce the noise encountered during driving. Such noise includes road conditions, driving maneuvers, vehicle instabilities, and mechanical vibrations. Since the quasi-static state of the vehicle typically lasts less than 30 s, a rapid algorithm is required to detect the driver’s HR for an initial assessment of their state. This initial assessment can also help in identifying false positives during HRV monitoring in the driving phase.

To address these challenges, this paper proposes advanced signal processing methods that combine the autocorrelation algorithm and the Huber–Kalman algorithm to rapidly and accurately acquire HR and HRV. The Huber–Kalman filtering method is introduced to effectively separate HRV from respiration [23,24], forming the basis for HRV measurement when the vehicle is stationary. This is the first application of the Huber–Kalman method in noncontact HRV monitoring. Additionally, the autocorrelation algorithm is employed to determine HR, enhancing the accuracy of HRV detection and filtering out anomalies detected by the Huber–Kalman algorithm. These methods are validated using a 60 GHz FMCW radar positioned behind the driver’s seat, demonstrating their efficacy in obtaining reliable HRV measurements during stops (e.g., at traffic lights) and smooth driving conditions (e.g., on highways).

The structure of the paper is as follows: Section 2 discusses the physiological basis and importance of HRV monitoring, providing an understanding of HRV, its regulation by the autonomic nervous system (ANS), and its significance in detecting stress, fatigue, and cardiovascular health. Section 3 explores the technological principles and applications of FMCW radar, focusing on its use for noncontact HRV monitoring. Section 4 delves into the advanced signal processing methods, including the classification of noise, the Huber–Kalman filtering method, and the short-window autocorrelation algorithm. Section 5 presents the experiments and results, describing the experimental setup, data analysis, and the accuracy evaluation of HR and HRV measurements. Finally, the findings and their implications are summarized and discussed for improving road safety in Section 6.

## 2. Physiological Basis and Importance of HRV Monitoring

This section explores the physiological mechanisms of HRV and its significance. We will define HRV, discuss its regulation by the ANS, and highlight its importance in detecting stress, fatigue, and cardiovascular health. Understanding these aspects underscores the critical role of HRV monitoring in ensuring driver safety.

### 2.1. Understanding HRV: Definition and Mechanisms

HRV denotes the fluctuations in the intervals between successive heartbeats. These time intervals are often measured as the R-R intervals (RRI) on an electrocardiogram (ECG). The RRI is the time between two successive R-waves, which are the peaks that represent the depolarization of the ventricles, the main pumping chambers of the heart. This interval is crucial because it reflects the heart’s ability to respond to various physiological conditions, including stress, exercise, and rest.

The ANS plays a significant role in regulating HRV. The ANS consists of two primary branches: the sympathetic and parasympathetic nervous systems. The sympathetic branch is responsible for the body’s fight or flight responses, causing an increase in HR and energy mobilization in response to stress or danger. On the other hand, the parasympathetic branch controls rest and digest functions, promoting relaxation and recovery by decreasing HR.

### 2.2. Clinical and Practical Importance of HRV

The importance of HRV lies in its ability to serve as a comprehensive indicator of the body’s cardiovascular health and autonomic function. Higher HRV is typically associated with a greater ability to adapt to stress and better cardiovascular fitness, suggesting a well-functioning ANS. Conversely, lower HRV can indicate stress, fatigue, or underlying health problems such as cardiovascular diseases.

Stress and Fatigue Detection: HRV monitoring can help in identifying stress and fatigue levels in individuals. For instance, a significant reduction in HRV can be an early indicator of mental or physical stress. This is particularly useful for drivers, where high stress or fatigue can lead to impaired driving performance and an increased risk of accidents.

Cardiovascular Health: HRV is a valuable marker in assessing the risk of cardiovascular diseases. Studies have shown that individuals with consistently low HRV are at a higher risk of developing conditions like hypertension, heart failure, and myocardial infarction. Continuous HRV monitoring can aid in the early detection and prevention of such diseases.

Drowsiness Detection: In driving scenarios, drowsiness detection is critical for preventing accidents. HRV can provide real-time insights into a driver’s alertness level. A drop in HRV might indicate the onset of drowsiness, prompting timely interventions such as alarms or automated driving assistance.

## 3. Technological Principles and Applications of FMCW Radar

Building on the physiological importance and monitoring significance of HRV discussed earlier, this section explores the technological methods for HRV detection. We focus on FMCW radar technology, a noncontact, highly sensitive solution operating at mmWave frequencies. By understanding how mmWave radar functions and its integration into vehicles, we can appreciate its crucial role in enhancing driver safety through accurate HRV monitoring.

### 3.1. Overview of Millimeter-Wave Radar

The mmWave radar operates at extremely high frequencies, typically between 30 GHz and 300 GHz, allowing it to achieve high resolution and sensitivity. These characteristics make mmWave radar ideal for detecting small movements such as those caused by human respiration and heartbeats. The short wavelength of mmWave radar enables it to detect very fine movements, such as the minute chest displacements caused by heartbeats and breathing, making it an ideal system for non-intrusive and contactless HRV monitoring. Moreover, mmWave radar can operate effectively under various environmental conditions, including darkness and through clothing, ensuring reliable performance in diverse scenarios, which is particularly suitable for automotive applications.

The compact size and versatility of mmWave radar make it ideal for integration into vehicles. Typically, the radar sensor is mounted behind the driver’s seat, with the antenna array aimed at the driver’s back, specifically targeting the heart region. This strategic placement offers several advantages: it ensures close proximity to the heart for precise detection, provides a noncontact method for continuous health monitoring without disrupting the driver’s comfort, and utilizes a stable mounting surface that minimizes the impact of vibrations and movements on radar accuracy. This setup is illustrated in Figure 1.

### 3.2. FMCW Radar: Basic Operating Principles

FMCW radar is a type of radar system that emits a continuous signal whose frequency is modulated over time. This signal, often referred to as a chirp, increases linearly in frequency over a specified period. The basic operating principles of FMCW radar include range measurement, velocity detection, and vital sign monitoring.

The hardware selected for this project is the Texas Instruments Incorporated (Dallas, TX, USA) AWR6843AOP, a single-chip 60 GHz mmWave sensor [25]. This sensor is specifically designed for low-power, high-accuracy applications such as driver vital sign monitoring. The 60 GHz frequency band is legally authorized in many regions, including the United States and Europe.

Chirp Signal Transmission:

FMCW radar transmits a chirp signal that sweeps linearly across a range of frequencies. When this signal encounters an object, it reflects back to the radar system.

Time Delay Measurement:

The expression of the transmission signal of the continuous frequency-modulated wave shown in Figure 2 is as follows [26]:(1)$$s_{TX}\left(t\right)=A_{TX}\cos\;\left[\int \left(w_{0}+A_{b}*t\right)dt\right]=A_{TX}\cos\;\left[(w_{0}+\frac{A_{b}}{2}*t)*t\right]$$

Upon a delay of $T_{d}$, the signal reflected from the object is captured. The received signal is expressed as:(2)$$s_{RX}\left(t\right)=\beta *s_{TX}\left(t-T_{d}\right)=A_{RX}\cos\;\left[(w_{0}+\frac{A_{b}}{2}*\left(t-T_{d}\right))*\left(t-T_{d}\right)\right]$$
β represents the attenuation factor encountered by the signal during propagation. Therefore, the distance between the object and the radar can be calculated by the time difference between the transmitted signal and the received signal. However, it is difficult to obtain the time delay directly. Therefore, we calculate the time by the frequency difference between the transmitted signal and the received signal.

Frequency Difference Calculation:

By mixing the received signal with a portion of the transmitted signal, an intermediate frequency (IF) signal is generated. This IF signal’s frequency is indicative of the object’s range. Applying a Fast Fourier Transform (FFT) to the IF signal allows the system to determine the range of the object accurately. The signal formula after the mixer is [26]:(3)$$\begin{array}{ll} s_{IF}\left(t\right) & =s_{RX}\left(t\right)*s_{TX}\left(t\right)\\ & \;=\frac{A_{TX}*A_{RX}}{2}[\cos(2*w_{0}*t+A_{b}*t^{2}-A_{b}*T_{d}*t+\frac{A_{b}}{2}*T_{d}^{2}\\ & \;-w_{0}*T_{d})+\cos\left(A_{b}*T_{d}*t+w_{0}*T_{d}-\frac{A_{b}}{2}*T_{d}^{2}\right)] \end{array}$$

The first cosine function represents the high-frequency component, which is filtered by a low pass filter, and the second cosine function is the low-frequency component, describing a beat signal at a fixed frequency. In $T_{d}=2\;*\frac{R}{c}$, the R denotes the radar-to-target distance.

Data Acquisition:

The data acquisition process involves capturing the reflected radar signals from the driver’s back. These signals contain information about the chest movements due to heartbeats and respiration. The radar sensor continuously emits FMCW signals and records the IF signals that result from the mixing of the transmitted and received signals.

Target Detection:

The first step in the signal processing workflow is detecting the presence of a target (i.e., the driver’s back). This is achieved using the radar’s ability to measure distance and detect movement within a predefined range.

Distance Measurement:

Once a target is detected, the radar measures the distance to the target by analyzing the frequency of the IF signal. The frequency difference $f_{IF}$ between the transmitted and received signals is directly related to the distance *d*:(4)$$f_{IF}=S*T_{d}$$ where *S* is the frequency modulation slope, and $T_{d}$ is the time delay. The distance *d* can then be calculated using:(5)$$d=\frac{c*T_{d}}{2}$$
where *c* denotes the speed of light.

Having covered the basic principles of FMCW radar, we can now delve into its specific application for vital sign monitoring, focusing on how it detects the small chest displacements caused by respiration and heartbeats.

### 3.3. Principles of FMCW Radar for Vital Sign Monitoring

For vital sign monitoring, FMCW radar detects the small chest displacements caused by respiration and heartbeats. These displacements result in phase changes in the reflected signal. Figure 3 depicts the overall system workflow for vital sign monitoring using FMCW radar. The workflow includes the emission of FMCW signals, reception and processing of reflected signals, and the extraction of vital signs such as heart and respiration rates.

Phase Change Analysis:

The radar continuously monitors the reflected signal’s phase. The phase change is directly related to the displacement of the chest, allowing the radar to capture the breathing pattern and heartbeat. The IF signal for multiple reflected signals is described by [26]:(6)$$s_{IF}\left(t\right)=\sum_{k=0}A_{k}*\cos\;\left(2*\pi *f_{k}*t+\phi_{k}+noise\right)$$
where *k* indicates different reflected signals captured by the receiver, $A_{k}$ denotes the reflected energy level from each target, and $\phi_{k}$ is the phase difference between the transmitted and received signals.

The radar continuously monitors the phase variations in the IF signal. The phase *ϕ* of the received signal is given by:(7)$$\phi \left(t\right)=\frac{4\pi d(t)}{\lambda }$$
where *λ* is the radar signal’s wavelength and *d*(*t*) denotes the displacement of the chest over time.

A 1D-FFT is applied to the time-domain IF signal to extract phase information from the signal. The In-phase (I) and Quadrature (Q) parts of the 1D-FFT signal are then used to demodulate the phase information:(8)$$\phi \left(t\right)=\arctan\;(\frac{Q}{I})$$

Phase Unwrapping:

Phase unwrapping resolves phase ambiguities that arise due to the periodic nature of the phase signal [23]. It ensures a continuous phase signal by adding or subtracting 2π whenever a phase discontinuity more significant than π is detected. This process is crucial for accurately tracking the chest movements over time.

Phase unwrapping is essential because the radar signal’s phase is periodic and can wrap around, causing sudden jumps in the phase value. If not corrected, these discontinuities can lead to errors in measuring chest displacement and, consequently, HRV.

Feature Extraction:

After obtaining the correct phase information, the following features are extracted:Respiration Rate: The phase signal’s low-frequency components, corresponding to slower, more significant movements, determine the respiration rate. The breathing pattern can be extracted by identifying the periodicity in these components;HR: To detect the HR, the high-frequency components of the phase signal corresponding to rapid, small movements are isolated. This is achieved by filtering out the respiration signal and focusing on the minor amplitude variations corresponding to heartbeats;HRV: HRV is calculated by analyzing the variations in the time intervals between consecutive heartbeats. These intervals, derived from the demodulated phase signal, provide insights into the ANS’s regulation of the heart.

Following this detailed signal processing workflow, the system accurately monitors the driver’s vital signs, providing critical information for enhancing road safety through continuous health monitoring.

## 4. Advanced Signal Processing Methods

Building upon the principles and applications of FMCW radar technology discussed in Section 3, this delves into the advanced signal processing methods essential for fast and accurate HRV monitoring. The effectiveness of HRV detection relies not only on the high sensitivity of mmWave radar but also on sophisticated algorithms to filter out noise and accurately extract vital signals. Here, we will explore the classification of noise, the Huber–Kalman filtering method, and the short-window autocorrelation algorithm, all of which play a critical role in enhancing the reliability and accuracy of HRV monitoring in real-world conditions.

### 4.1. Classification of Noise and Challenges

#### 4.1.1. Interference from Respiration Signals

Respiration signals present a significant challenge in accurately detecting HR and HRV using mmWave radar. The primary issue arises from the fact that the chest movements caused by breathing are much larger than those caused by heartbeats. Respiration typically results in chest displacements of approximately 5 mm, whereas heartbeats cause displacements averaging around 0.5 mm.

Due to the relatively small displacement associated with heartbeats, the HRV signal is quickly overshadowed by the more pronounced movements from respiration. This overlap can lead to difficulty isolating the HRV signal from the respiratory signal. Advanced signal processing techniques, such as the Huber–Kalman filter, are necessary to differentiate between these two types of displacements. The filter effectively separates the smaller, HRV-related movements from the larger, respiration-induced displacements.

#### 4.1.2. Dynamic Noise in the Driving Environment

Another significant source of noise in HRV monitoring using mmWave radar comes from the dynamic conditions experienced during driving. The vehicle’s motion introduces various types of noise that can interfere with accurate HRV detection. These include the following:Road Conditions: Bumps, gravel, and potholes can cause abrupt movements of the driver’s body relative to the sensor, resulting in significant displacements ranging from millimeters to tens of centimeters;Driver’s Maneuvers: Actions such as hand movements, adjusting posture, and other body movements can also create noise that complicates HRV signal detection;Vehicle Dynamics: Acceleration, deceleration, and turning movements further contribute to the displacements between the driver’s body and the sensor. These movements are often random and unpredictable, making it challenging to filter out the noise effectively.

The combined effect of these factors results in a complex and unpredictable noise environment that can obscure the HRV signal. Despite efforts to apply advanced filtering techniques, such as the Huber–Kalman filter and short-windowed autocorrelation methods, the challenge remains substantial due to the inherent variability and randomness of driving-induced movements.

Given these challenges, it is recommended that HR and HRV be monitored primarily when the vehicle is stable, such as when it is parked or moving smoothly on well-maintained roads. During bumpy or irregular road conditions, it becomes exceedingly difficult to obtain accurate HRV measurements due to the excessive noise introduced by the factors mentioned above. Therefore, ensuring that HRV detection is performed during stable driving conditions will improve the reliability and accuracy of the data collected.

### 4.2. Huber–Kalman Filtering Method

The Kalman filter algorithm is highly effective for filtering phase signals to detect vital signs [23,27]. However, during driving, more significant disturbances occur due to vehicle motion, which requires optimization of the Kalman filter. This is where the Huber algorithm comes into play, enhancing the robustness of the Kalman filter by handling outliers and reducing the impact of significant disturbances. This optimized approach has proven effective when applied in Wi-Fi-based driver respiration detection [23]. When adapted for millimeter-wave radar, the combination of Kalman and Huber filtering offers superior performance, yielding more accurate HR and HRV measurements.

#### 4.2.1. Algorithm Principles

The Kalman filter is a highly effective data processing tool known for its real-time capabilities, accuracy, and speed. It is particularly adept at reducing measurement errors and managing random noise within a system. Kalman filters are widely used in various signal processing and control domains, including navigation, tracking, and communications. As a recursive algorithm, the Kalman filter does not require storing all past measurement data; it only uses current data for calculations, making it highly efficient for real-time applications. However, the Kalman filter is sensitive to outliers in the measurement data, which can increase estimation errors and potentially cause filter divergence. The Huber–Kalman filter was introduced to address the sensitivity of the Kalman filter to outliers. The Huber–Kalman filter incorporates the robustness of the Huber loss function into the Kalman filter framework, enhancing its resistance to outliers. The Huber loss function behaves like the squared loss for minor errors. Still, it transitions to linear growth for errors exceeding a certain threshold, thus reducing the impact of outliers on the estimation results. Specifically, the Huber–Kalman filter reweights the errors during the state update phase, reducing the weight of significant errors (potentially outliers) and mitigating their adverse effects on state estimation.
(9)$$\rho_{a}\left(f\right)=\left\{\begin{array}{ll} \frac{f^{2}}{2} & if\;f\leq a\\ a\left|f\right|-\frac{a^{2}}{2} & if\;f>a \end{array}\right.$$
let a be a positive constant that defines the boundary between the significant and minor errors and let f represent the error function. By adjusting a, the balance between the norms $l_{1}$ or $l_{2}$ in processing error functions can be controlled. When a is large, the optimization problem aligns with the traditional least squares approach. Conversely, as a approaches zero, the optimization problem shifts towards a first norm-based method.

The prior estimation error $e_{k}^{-}$ and the posterior estimation error $e_{k}$ represent the difference between the prior estimate ${\tilde{x}}_{k}^{-}$, the optimal estimate ${\tilde{x}}_{k}$, and the actual value $x_{k}$, respectively. The covariances of these estimation errors are described by the following functions:(10)$$P_{k}^{-}=\left\{\begin{array}{ll} \frac{e_{k}^{-2}}{2} & if\left|e_{k}^{-}\right|\leq a\;\\ a\left|e_{k}^{-}\right|-\frac{a^{2}}{2} & if\;\left|e_{k}^{-}\right|>a \end{array}\right.$$
(11)$$P_{k}=\left\{\begin{array}{ll} \frac{e_{k}^{2}}{2} & if\left|e_{k}\right|\leq a\;\\ a\left|e_{k}\right|-\frac{a^{2}}{2} & if\;\left|e_{k}\right|>a \end{array}\right.$$

Combined ${\tilde{x}}_{k}={\tilde{x}}_{k}^{-}+K_{k}\left(z_{k}-H{\tilde{x}}_{k}^{-}\right)$, $P_{k}$ represented by $P_{k}^{-}$ as
(12)$$P_{k}=\left\{\begin{array}{c} \frac{\left(I-KH\right)P_{k}^{-}{\left(I-KH\right)}^{T}+KRK^{T}}{2}\;if\;0\leq P_{k}^{-}\leq \frac{a^{2}}{2}\\ \left(I-KH\right)\left(P_{k}^{-}+\frac{a^{2}}{2}\right)-\frac{a^{2}}{2}\;if\;P_{k}^{-}>\frac{a^{2}}{2} \end{array}\right.$$

The Kalman filter’s estimation method aims to minimize the error in the state estimation function to closely match the actual value.

Taking the partial derivative of Equation (13) with respect to the Kalman gain coefficient K results in:(13)$$\frac{\partial P_{k}}{\partial K_{k}}=\left\{\begin{array}{c} -P_{k}^{-}H^{T}+K\left(HP_{k}^{-}H^{T}+R\right)\\ if\;0\leq P_{k}^{-}\leq \frac{a^{2}}{2}\\ \frac{HK_{k}H^{T}-H}{\left|1-K_{k}H\right|}\left(P_{k}^{-}+\frac{a^{2}}{2}\;\right)\\ \;if\;P_{k}^{-}>\frac{a^{2}}{2} \end{array}\right.$$

By setting the partial derivative in Equation (14) to zero, we obtain
(14)$$K_{k}=\left\{\begin{array}{ll} \frac{P_{k}^{-}H^{T}}{HP_{k}^{-}H^{T}+R} & if\;0\leq \;P_{k}^{-}\leq \frac{a^{2}}{2}\;\\ \frac{1}{H} & if\;P_{k}^{-}>\frac{a^{2}}{2} \end{array}\right.$$

It is clear that ${\;P}_{k}^{-}$ can be represented using $P_{k-1}$ as follows:(15)$$P_{k}^{-}=\left\{\begin{array}{c} \frac{AP_{k-1}A^{T}+Q}{2}\;if\;0\leq \;P_{k-1}\leq \frac{a^{2}}{2}\\ \left|A\right|\left(P_{k-1}+\frac{a^{2}}{2}\;\right)-\frac{a^{2}}{2}\;if\;P_{k-1}>\frac{a^{2}}{2} \end{array}\right.$$

Figure 4 presents the prediction-update model of the Huber–Kalman filter. This model illustrates how the Huber–Kalman filter improves upon the traditional Kalman filter by incorporating the Huber loss function, which enhances the filter’s robustness against outliers and significant disturbances. The model outlines the state prediction and correction process, highlighting the integration of Huber weighting to mitigate the impact of measurement errors and ensure more accurate state estimation in dynamic environments.

#### 4.2.2. HRV Measurement via Huber–Kalman Filter

As mentioned above, the IF signal phase of FMCW radar reflects the displacement of the chest. This displacement encompasses both respiratory and heartbeat movements. The displacement caused by heartbeat is significantly smaller than that caused by respiration. Heartbeat-induced chest movements typically range from 0.2 mm to 0.6 mm, whereas respiratory movements can cause chest displacements of 4 mm to 12 mm. The amplitude of heartbeat-induced chest displacement is significantly weaker than that of respiratory movements, which increases the difficulty of separating respiratory and heartbeat signals. However, respiration and heartbeat exhibit different patterns. A smooth, rhythmic pattern characterizes respiratory movements; in contrast, the heartbeat movements are rapid and follow a sudden motion pattern, with each beat causing a quick, sharp displacement followed by a brief pause. This periodic nature of respiration contrasts with the intermittent, sudden motions of heartbeat-induced chest movements. Given the direct correlation between phase and chest displacement, the derivative of the phase can be used to determine the velocity of chest displacement. By calculating the velocity, it is possible to enhance the heartbeat signal, which facilitates the accurate calculation of HRV.

We can develop a model utilizing the Huber–Kalman filter to estimate the velocity of chest displacement. The Huber–Kalman filter is particularly effective in dealing with non-Gaussian noise and outliers, enhancing robustness and accuracy in state estimation. By applying this filter, we can accurately determine the velocity of chest movements, improving the heartbeat signal. Detecting HRV requires accurate RRI, which is more challenging than regular HR detection. Standard methods, such as band-pass filtering or spectral analysis, are unsuitable for HRV detection. Because band-pass filters smooth the waveform, accurate RRI cannot be obtained. The Huber–Kalman algorithm enhances the characteristics of the R-wave, significantly improving the accuracy of HRV detection. It effectively separates HRV signals from respiratory signals. This improvement is crucial for the precise calculation of HRV.

We first build the physical model. In this model, the observation variable is the chest displacement z, and the state variables include both displacement ${\tilde{x}}_{k}$ and velocity ${\tilde{v}}_{k}$. The system can be described using the following state-space equations:

State Equation: This describes the evolution of the system state over time. For displacement and velocity, the state transition model is given by
(16)$$\left(\begin{array}{c} {\tilde{x}}_{k+1}\\ {\tilde{v}}_{k+1} \end{array}\right)=\left(\begin{array}{cc} 1 & ∆t\\ 0 & 1 \end{array}\right)\left(\begin{array}{c} {\tilde{x}}_{k}\\ {\tilde{v}}_{k} \end{array}\right)+\left(\begin{array}{c} \frac{1}{2}∆t^{2}\\ ∆t \end{array}\right)u_{k}+\omega_{k}$$

Here, ${\tilde{x}}_{k}$ is the displacement at time k, ${\tilde{v}}_{k}$ is the velocity of the chest movement, Δt is the time step, $u_{k}$ is the acceleration, and $\omega_{k}$ is the process noise.

Observation Equation: This relates the observed displacement to the state variables:(17)$$z_{k}=x_{k}+n_{k}$$

Here, $z_{k}$ is the observed displacement and $n_{k}$ is the measurement noise.

By incorporating these equations, the Kalman filter can iteratively estimate the state vector $\left(\begin{array}{c} {\tilde{x}}_{k}\\ {\tilde{v}}_{k} \end{array}\right)$, providing robust velocity estimations even in the presence of noise. This model forms the basis for further applications, such as enhancing heartbeat signal detection and HRV analysis.

Figure 5 illustrates the impact of body movements on radar phase signals and compares the performance of the Kalman filter and the Huber–Kalman filter in mitigating these effects. The top subplot shows the raw radar phase signal with noticeable disturbances caused by intentional body movements, highlighted in red boxes. These body movements were simulated in a stationary and turned-off vehicle, with the radar installed behind the car seat (as shown in Figure 1). The bottom subplot compares the RRI calculated using the Polar H10 (reference device), Kalman, and Huber–Kalman filters. The Huber–Kalman filter (red line) provides a more consistent and accurate RRI measurement compared to the Kalman filter (blue line), closely matching the reference data (green line), thereby demonstrating its effectiveness in handling noise and body movements for reliable HRV monitoring.

From Figure 6, the three subplots in the figure respectively display the raw radar phase signal, the heartbeat signal after Huber–Kalman filtering, and the comparison of the interval between successive heartbeats (RRI) calculated by the Huber–Kalman filter and the reference device Polar H10, which is a wearable golden standard. The second subplot presents the signal after being processed by the Huber–Kalman filter. This filtered signal prominently highlights the heartbeat positions (marked in red). Compared to the raw radar phase in the first subplot, the processed signal clearly shows the heartbeat locations, facilitating accurate heartbeat signal extraction. The third subplot compares the RRIs calculated by the radar after Huber–Kalman filtering and those measured by the reference device, Polar H10. The plot indicates that the RRI trend obtained from the Huber–Kalman filtered signal closely matches the measurements from the Polar H10, demonstrating high consistency.

In conclusion, it is evident that after applying the Huber–Kalman filter with this model, the velocity changes caused by the heartbeat are significantly amplified, creating distinct peaks. These peaks enable the effective capture of heartbeat timestamps, allowing for the calculation of RRI and HRV.

### 4.3. Short-Window Autocorrelation Algorithm

While the Huber–Kalman algorithm is effective, it can only sometimes ensure accurate HRV results, particularly when phase signal quality is compromised, such as on bumpy roads. Additionally, the Huber–Kalman algorithm does not assess the reliability of HRV results. Therefore, an additional algorithm is necessary for post-processing analysis.

The short-window autocorrelation algorithm is essential for rapidly extracting HR by leveraging the periodic characteristics of physiological signals. By implementing a moving autocorrelation sliding window, the algorithm estimates the periodicity of heartbeats, allowing for continuous and unobtrusive cardiac monitoring. This capability is precious in quick and reliable cardiac monitoring scenarios.

The autocorrelation function measures the similarity between a signal and a delayed version of itself over various time lags, identifying periodic patterns that correspond to heartbeats. This method is effective for short-term HR estimation because it can rapidly determine the fundamental frequency of the heartbeat signal within a short observation window, thereby providing real-time and robust HR measurements.

The autocorrelation function $r_{k}$ for the chosen radar phase time series *y* at lag *k* is computed as follows:(18)$$r_{k}=\frac{\frac{1}{T}\sum_{t=1}^{T-k}\left(y_{t}-\bar{y})(y_{t+k}-\bar{y}\right)}{c_{0}}$$
where T is the total number of time series data points, $\bar{y}$ represents the mean of y, and $c_{0}$ is the sample variance of y. The periodic lag $k_{m}$ corresponding to the frequency range of 0.6–2 Hz is then identified by locating the maximum within this range $r_{k}$. This step can be straightforwardly implemented. Consequently, the HR can be estimated as follows:(19)$$f_{HR}=\frac{1}{k_{m}∆t}$$

By inputting 3 s of radar phase data, if the autocorrelation result $r_{k}$ shows a prominent peak, it indicates a high confidence level in the current HR. Conversely, a less pronounced peak suggests significant interference, and the current HR should not be used to remove anomalous RRI. When a high-confidence HR is present, the acceptable range for RRI should be calculated as $\frac{60,000}{HR}\pm 200\;\mathrm{ms}$. Any RRI outside this range should be discarded.

As shown in Figure 7, a 3 s time window was selected and processed through a band-pass filter to retain only the high-frequency micro-features. The autocorrelation algorithm was then applied, yielding an HR result of 70 bpm, compared to the reference device, which showed an actual HR of 71 bpm. The comparison between the autocorrelation and FFT results indicates that the autocorrelation result is closer to the exact value. Because the spectral resolution of FFT is limited and requires a more extended sampling period for accurate results. However, the autocorrelation algorithm proves to be more effective for short-term HR detection and tracking HR changes.

Additionally, the autocorrelation results can be used to evaluate signal quality. When the signal quality is good, the HR results from the autocorrelation are prominent. Conversely, the HR results could be more prominent and attainable when the signal quality is poor. If the HR results cannot be obtained, it is impossible to determine HRV.

## 5. Experiments and Results

Building on the theoretical foundations and advanced signal processing methods discussed in the previous sections, this section delves into the practical application and evaluation of the proposed wireless intelligent sensor system. We conducted experiments to assess the system’s performance in real-world scenarios. This section will describe the experimental setup, present the collected data, and analyze the results to demonstrate the system’s effectiveness and accuracy in monitoring driver HRV.

### 5.1. Experimental Setup

The experiments were conducted with five participants to evaluate the performance of the proposed wireless intelligent sensor system for monitoring HRV in drivers. These participants were tested across two vehicle models: Mode V and Model S. Figure 8 shows the 60 GHz FMCW radar system installation behind the driver’s seat. This strategic placement ensures optimal detection of the driver’s HR and respiration by providing a stable, unobstructed line of sight to the driver’s chest area, thereby enhancing the accuracy and reliability of the health monitoring system in real-world driving conditions.

### 5.2. Data Results Analysis

The data analysis focused on evaluating the accuracy of HR and HRV measurements under various conditions. Table 1 summarizes the results obtained from different vehicle models and test persons, including the average and HRV errors.

Table 1 summarizes HR and HRV measurement errors across vehicle models and test subjects using the proposed wireless intelligent sensor system. The system’s HRV detection employs a 60 GHz FMCW radar and advanced signal processing algorithms. The results show that the average HR error across all participants was approximately 0.34 bpm, and the average HRV (Mean RRI) error was about 2.21 ms. These low error margins demonstrate the system’s high accuracy and reliability in real-world conditions. The effectiveness of the Huber–Kalman filtering and autocorrelation algorithms is evident, as they successfully mitigate noise and accurately capture HR and HRV data. After thorough testing, we confirmed that the type of vehicle did not have a significant effect on the experimental results. Our system demonstrated consistent performance across different car models. According to the Task Force of the European Society of Cardiology and the North American Society of Pacing and Electrophysiology, HRV standards specify that a measurement error of less than 5% is acceptable for time-domain methods such as SDNN and RMSSD [28]. These guidelines are widely recognized and serve as a benchmark for HRV monitoring systems. Our system’s HRV testing error is less than 5%.

The accuracy of HRV measurements is further supported by analyzing specific HRV metrics such as SDNN (Standard Deviation of NN intervals) [5,9] and RMSSD (Root Mean Square of Successive Differences) [2,5]. SDNN and RMSSD are crucial metrics in the analysis of HRV. SDNN measures the variability in the time intervals between RRI over a given period, providing an overall assessment of ANS activity and reflecting the balance between sympathetic and parasympathetic nervous system inputs. RMSSD, on the other hand, focuses on the short-term components of HRV by calculating the square root of the mean of the squares of successive differences between adjacent RRI. This metric is particularly sensitive to high-frequency HR variations, primarily influenced by parasympathetic activity. Together, SDNN and RMSSD offer comprehensive insights into cardiovascular health, stress levels, and autonomic regulation, making them essential for evaluating the effectiveness of HRV monitoring systems.

Table 2 compares HRV time-domain results between the radar system and the reference device, Polar H10. The table includes critical metrics such as Mean RRI, Mean HR, Min HR, Max HR, SDNN, and RMSSD across different test participants. The radar system’s SDNN values showed an average error range of 0.6 ms to 4.2 ms compared to the Polar H10. RMSSD focuses on short-term HRV by calculating the square root of the mean of the squares of successive differences between adjacent NN intervals. The radar system’s RMSSD values had an average error range of 1.9 ms to 12.6 ms compared to the Polar H10. This higher sensitivity to short-term HRV changes is beneficial for detecting rapid fluctuations in HR, indicative of parasympathetic nervous system activity.

## 6. Conclusions

In this study, we addressed several challenging problems associated with noncontact HRV monitoring, particularly the need for rapid and accurate detection. Traditional methods often require 30 s or more to achieve reliable measurements, which is impractical for real-time driver monitoring due to the potential for variable data quality over extended periods. Quick detection is crucial in driver monitoring scenarios, as data quality cannot be guaranteed to remain reasonably good for long durations.

Our original contributions to solving these problems are significant as follows: (1) we can claim that it is the first time to apply the Huber–Kalman algorithm in HRV monitoring; (2) the autocorrelation algorithm is used to swiftly determine HR within 3 s and assess signal quality based on the prominence of peaks in the autocorrelation results; (3) the Huber–Kalman method may include some erroneous RRI results, but these can be filtered out using the HR results obtained from the autocorrelation analysis. Therefore, by combining the Huber–Kalman and autocorrelation algorithms, we have achieved fast and accurate HRV monitoring; and (4) the results demonstrate high precision, with the SDNN error within 5 ms, RMSSD error within 13 ms, and Mean RRI error within 11 ms.

Despite these advancements, several challenges remain for future research. Achieving stable HRV monitoring during vehicle movement has not been fully realized, necessitating further investigation. Ensuring reliable HRV measurements in dynamic driving conditions is crucial for enhancing the robustness and applicability of our system in real-world scenarios. Future efforts will focus on optimizing signal processing algorithms to improve performance in more dynamic and noisy environments. The ultimate goal is to seamlessly integrate our HRV monitoring system into driver assistance technologies, thereby enhancing road safety and health monitoring.

## Figures and Tables

**Figure 1 biomimetics-09-00481-f001:**
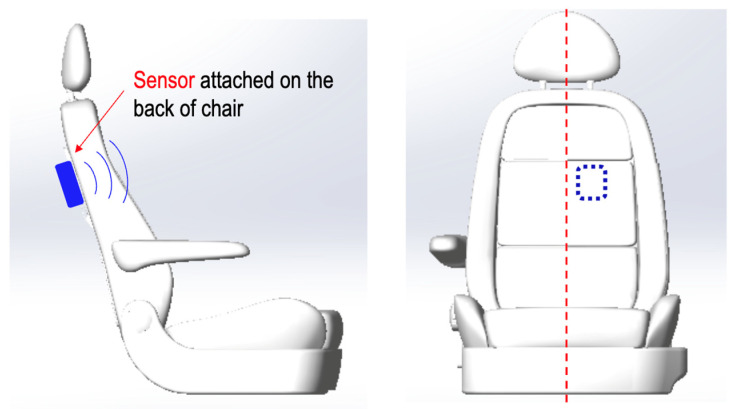
Installation location.

**Figure 2 biomimetics-09-00481-f002:**
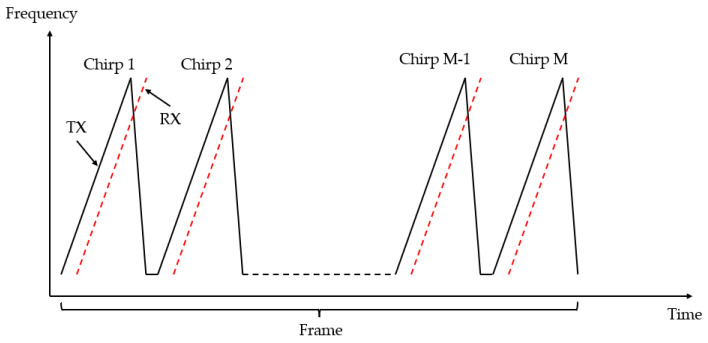
Transmitted and received frequency-modulated signals.

**Figure 3 biomimetics-09-00481-f003:**
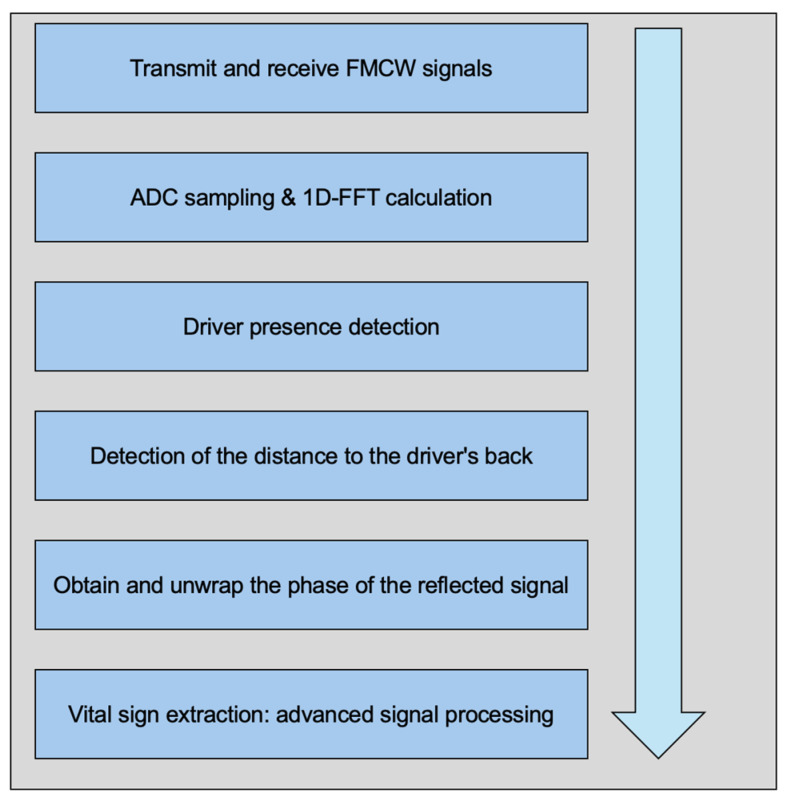
Overall system workflow.

**Figure 4 biomimetics-09-00481-f004:**
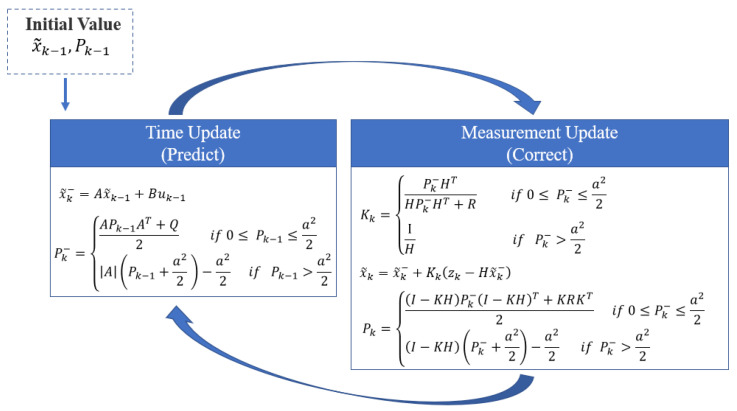
Huber–Kalman filter prediction-update model.

**Figure 5 biomimetics-09-00481-f005:**
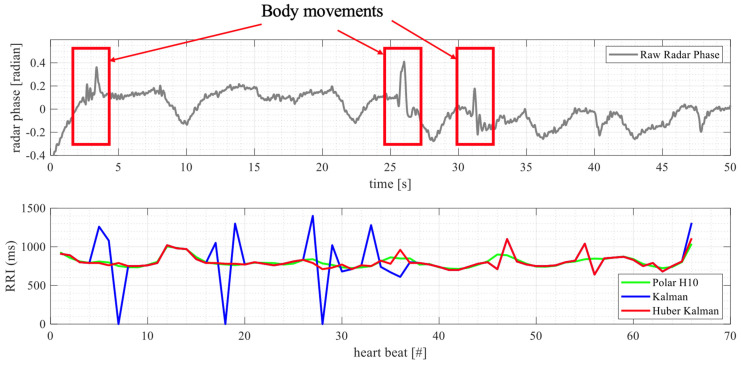
Comparison of Kalman filter and Huber–Kalman filter.

**Figure 6 biomimetics-09-00481-f006:**
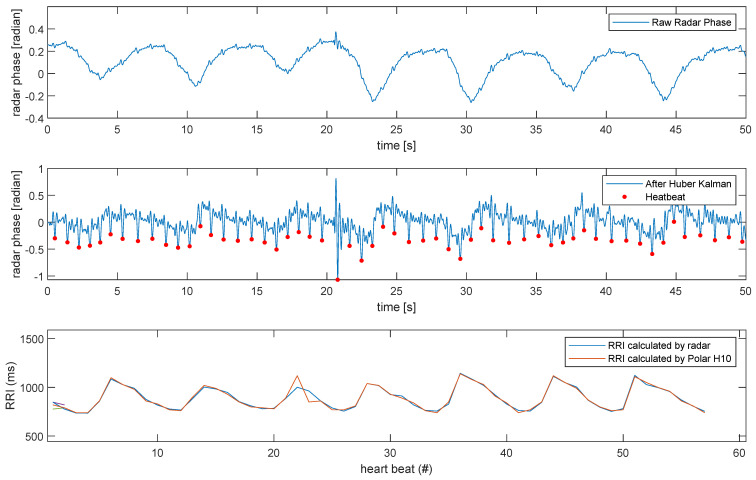
Effectiveness of Huber–Kalman filter in heartbeat signal extraction.

**Figure 7 biomimetics-09-00481-f007:**
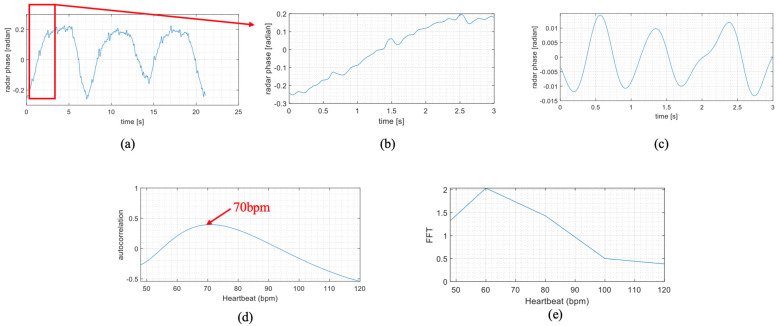
Fast HR detection based on autocorrelation algorithm: (**a**) Raw Vital Sign Data from Radar; (**b**) 3-Second Segment of Raw Data; (**c**) Band-Pass Filtered 3-Second Data; (**d**) HR Calculation Using Autocorrelation; (**e**) HR Calculation Using FFT.

**Figure 8 biomimetics-09-00481-f008:**
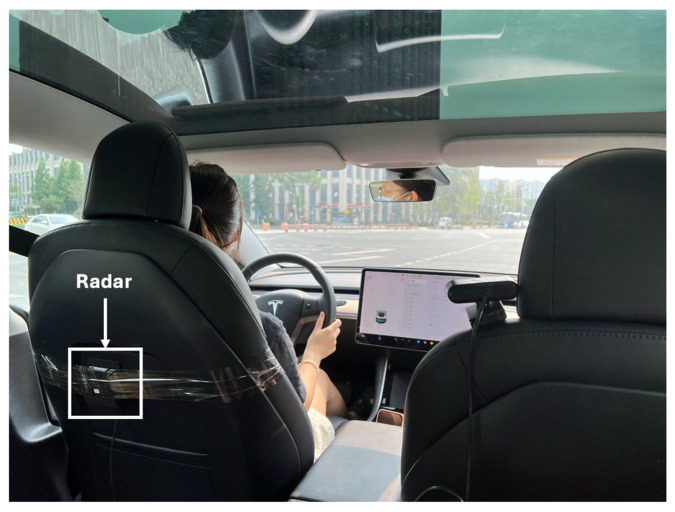
Installation of the 60 GHz FMCW radar system behind the driver’s seat.

**Table 1 biomimetics-09-00481-t001:** Summary of HR and HRV measurement errors across different vehicle models and test persons.

Vehicle Model	Test Person ID	HR (bpm)			MEAN RRI (ms)		
		This Work	Polar H10	Error	This Work	Polar H10	Error
Model V	1	84.9	86.2	−1.3	706.18	696.08	10.1
	2	112.0	112.0	0	537.00	537.00	0
	3	90.4	90.4	0	663.97	663.94	0.03
Model S	4	107.7	108	−0.3	556.94	557.00	−0.06
	5	78.4	78.3	0.1	765.01	765.89	−0.88
**Average Error**				**0.34**			**2.21**

**Table 2 biomimetics-09-00481-t002:** HRV time-domain analysis results.

Test Person ID	Variable	Units	Value		
			This Work	Polar H10	Error
**1**	**Mean RRI**	**ms**	706.18	696.08	10.1
	**Mean HR**	**beats/min**	84.964	86.197	−1.233
	**Min HR**	**beats/min**	74.184	76.766	−2.582
	**Max HR**	**beats/min**	96.277	95.178	1.099
	**SDNN**	**ms**	36.307	32.764	3.543
	**RMSSD**	**ms**	33.060	20.494	12.566
**2**	**Mean RRI**	**ms**	537	537	0
	**Mean HR**	**beats/min**	112	112	0
	**Min HR**	**beats/min**	108	107	1
	**Max HR**	**beats/min**	118	117	1
	**SDNN**	**ms**	8.7	9.3	−0.6
	**RMSSD**	**ms**	9.8	11.7	−1.9
**3**	**Mean RRI**	**ms**	663.97	663.94	0.03
	**Mean HR**	**beats/min**	90.366	90.369	−0.003
	**Min HR**	**beats/min**	73.457	73.135	0.322
	**Max HR**	**beats/min**	97.911	98.361	−0.45
	**SDNN**	**ms**	36.105	31.908	4.197
	**RMSSD**	**ms**	40.680	30.999	9.681
**4**	**Mean RRI**	**ms**	556.94	557	−0.06
	**Mean HR**	**beats/min**	107.73	108	−0.27
	**Min HR**	**beats/min**	98.555	99	−0.445
	**Max HR**	**beats/min**	113.12	113	0.12
	**SDNN**	**ms**	17.773	14.5	3.273
	**RMSSD**	**ms**	21.373	9.5	11.873
**5**	**Mean RRI**	**ms**	765.01	765.89	−0.88
	**Mean HR**	**beats/min**	78.430	78.341	0.089
	**Min HR**	**beats/min**	74.111	74.004	0.107
	**Max HR**	**beats/min**	82.873	83.156	−0.283
	**SDNN**	**ms**	17.226	15.719	1.507
	**RMSSD**	**ms**	22.488	18.596	3.892

## Data Availability

Data are contained within the article.

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
