# Peer review of "Enhancing Road Safety: Fast and Accurate Noncontact Driver HRV Detection Based on Huber–Kalman and Autocorrelation Algorithms"

_biomimetics, 2024, doi:10.3390/biomimetics9080481_

Round 1
Reviewer 1 Report
Comments and Suggestions for Authors
In Figure 5, the situation setting was not readable. Specifically, it was not explicitly stated whether the red box in the top subplot was the effect of the intended human body motion or other noise from vehicle dynamics or other sources.
Author Response
Comments 1: In Figure 5, the situation setting was not readable. Specifically, it was not explicitly stated whether the red box in the top subplot was the effect of the intended human body motion or other noise from vehicle dynamics or other sources.
Response 1: Thank you for your thorough review and valuable feedback on our paper.
The noise highlighted in the red boxes is indeed due to intentional body movements. To simulate this, the radar was installed behind the car seat (as shown in Figure 8), and the tests were conducted with the vehicle stationary and turned off. This ensures that all detected disturbances are solely from human body movements and not from vehicle dynamics or other external noise sources.
Change: Revised the paragraph in 4.2.2.
Figure 5 illustrates the impact of body movements on radar phase signals and compares the performance of the Kalman filter and the Huber-Kalman filter in mitigating these effects. The top subplot shows the raw radar phase signal with noticeable disturbances caused by intentional body movements, highlighted in red boxes. These body movements were simulated in a stationary and turned-off vehicle, with the radar installed behind the car seat (as shown in Figure 8). The bottom subplot compares the RRI calculated using the Polar H10 (reference device), Kalman, and Huber-Kalman filters. The Huber-Kalman filter (red line) provides a more consistent and accurate RRI measurement compared to the Kalman filter (blue line), closely matching the reference data (green line), thereby demonstrating its effectiveness in handling noise and body movements for reliable HRV monitoring.

Reviewer 2 Report
Comments and Suggestions for Authors
This work focuses on a wireless intelligent sensor 11 system that measures the heart rate variability of 12 drivers using millimeter-wave radar. This article primarily explains the use of sophisticated signal processing techniques, such as the Huber-Kalman filtering algorithm, to lessen the effect of breathing on heart rate detection. However, there are some concerns need to be addressed before considering it for publication.
1. Since the Huber Kalman filter is used to measure HRV, that is the main concept. The authors need to assess the filter's capacity to manage noise and anomalies in the identification of heart rate variability. Furthermore, the authors need to investigate into What is its performance in contrast to other filters?
2. It was mentioned “the Huber-Kalman algorithm does not assess the reliability of HRV results. Therefore, an additional algorithm is necessary for post-processing analysis. The short-window autocorrelation algorithm is essential for rapidly extracting HR 439 by leveraging the periodic characteristics of physiological signals”. How the author have chosen this algorithm? Whether the author have compared this one with any other algorithms? This should be clarified
3. Tables 1 and 2 show that there are a few inaccuracies. In what way are these mistakes analyzed? The author ought to make an effort to look into the error margins mentioned in the literature. Is it appropriate for the author to discuss if the false positive and false negative rates are acceptable for real-world applications?
4. The author should make sure that a detailed comparison with current HRV detection technology is included in the study. In what ways does the suggested approach improve upon or supplement these technologies?
5. The paper does not examine the constraints of the proposed technique. The restrictions need to be addressed and explicitly communicated.
6. “These participants were tested across two vehicle models: Mode V and Model S” Why it is tested with two different model. Is your method depends on vehicle or which result depends on the vehicle model? The authors should comment on this.
7. Apart from FMCW radar, what are the hardware design for this proposal? What is the standards used to make that hardware?
Comments on the Quality of English LanguageMinor editing is required
Author Response
Comments 1: Since the Huber Kalman filter is used to measure HRV, that is the main concept. The authors need to assess the filter's capacity to manage noise and anomalies in the identification of heart rate variability. Furthermore, the authors need to investigate into What is its performance in contrast to other filters?
Response 1: Thank you for your valuable comments.
Yes, HRV detection is one of the key aspects of this paper. We employed the Huber Kalman algorithm, an enhancement of the traditional Kalman filter, specifically designed to filter out anomalies caused by physical movements and other disturbances. The efficacy of this method is illustrated in Figure 5, which contrasts the traditional Kalman filter with the Huber Kalman filter, highlighting the latter's superior robustness and noise management.
Our results indicate that the Huber Kalman filter effectively mitigates the impact of respiration and other body movements, which are significant sources of noise in HRV measurements. The performance comparison shows that the Huber Kalman filter provides more consistent and accurate results, closely aligning with the reference data from the Polar H10 device, as detailed in the manuscript.
In addition, we analyzed the performance of our proposed filter against general bandpass filters (as shown in Figure 7). While bandpass filters can sometimes cause incomplete heart waveforms by filtering out essential high-frequency or low-frequency components, leading to significant HRV result errors, our Huber Kalman filter maintains the integrity of the RRI (R-R intervals). This is crucial because the variability in heartbeats is better captured in the time domain rather than the frequency domain, where bandpass filters are less effective.
Change: Revised the paragraph in 4.2.2.
Figure 5 illustrates the impact of body movements on radar phase signals and compares the performance of the Kalman filter and the Huber-Kalman filter in mitigating these effects. The top subplot shows the raw radar phase signal with noticeable disturbances caused by intentional body movements, highlighted in red boxes. These body movements were simulated in a stationary and turned-off vehicle, with the radar installed behind the car seat (as shown in Figure 8). The bottom subplot compares the RRI calculated using the Polar H10 (reference device), Kalman, and Huber-Kalman filters. The Huber-Kalman filter (red line) provides a more consistent and accurate RRI measurement compared to the Kalman filter (blue line), closely matching the reference data (green line), thereby demonstrating its effectiveness in handling noise and body movements for reliable HRV monitoring.
Comments 2: It was mentioned “the Huber-Kalman algorithm does not assess the reliability of HRV results. Therefore, an additional algorithm is necessary for post-processing analysis. The short-window autocorrelation algorithm is essential for rapidly extracting HR by leveraging the periodic characteristics of physiological signals”. How the author have chosen this algorithm? Whether the author have compared this one with any other algorithms? This should be clarified
Response 2: The necessity for rapid heart rate (HR) detection in driver monitoring scenarios is paramount due to the potential for heart rate variability within seconds. In such environments, an algorithm that can swiftly and accurately capture these changes is critical. We selected the short-window autocorrelation algorithm for its ability to detect accurate heart rates within a short 3-second window. This rapid detection is crucial in real-time monitoring systems where immediate responses to physiological changes are required.
The short-window autocorrelation algorithm was chosen after careful consideration of its strengths in time-domain analysis, which is particularly suited for dynamic and real-time applications. Compared to other algorithms such as the Fast Fourier Transform (FFT), the autocorrelation method demonstrated superior performance in terms of speed and accuracy within the constrained sampling period available in our application. FFT-based methods, while powerful for frequency domain analysis, require longer sampling periods to achieve comparable accuracy, as they are constrained by the need for sufficient frequency resolution. Shorter sampling times with FFT can result in significant detection errors, making them less suitable for our specific requirement of rapid and accurate HR detection.
Figure 7 in the manuscript illustrates this comparison, showing how the autocorrelation algorithm consistently yields accurate HR results even with shorter data windows. The comparison highlights the limitations of FFT in our application context, where the necessity for rapid detection often outweighs the benefits of frequency domain analysis.
Change: Revised the paragraph in 4.3.
The short-window autocorrelation algorithm is essential for rapidly extracting HR by leveraging the periodic characteristics of physiological signals. By implementing a moving autocorrelation sliding window, the algorithm estimates the periodicity of heartbeats, allowing for continuous and unobtrusive cardiac monitoring. This capability is precious in quick and reliable cardiac monitoring scenarios.
The autocorrelation function measures the similarity between a signal and a delayed version of itself over various time lags, identifying periodic patterns that correspond to heartbeats. This method is effective for short-term HR estimation because it can rapidly determine the fundamental frequency of the heartbeat signal within a short observation window, thereby providing real-time and robust HR measurements.
As shown in Figure 7, a 3-second time window was selected and processed through a band-pass filter to retain only the high-frequency micro-features. The autocorrelation algorithm was then applied, yielding an HR result of 70 bpm, compared to the reference device, which showed an actual HR of 71 bpm. The comparison between the autocor-relation and FFT results indicates that the autocorrelation result is closer to the exact value. Because the spectral resolution of FFT is limited and requires a more extended sampling period for accurate results. However, the autocorrelation algorithm proves to be more effective for short-term HR detection and tracking HR changes.
Comments 3: Tables 1 and 2 show that there are a few inaccuracies. In what way are these mistakes analyzed? The author ought to make an effort to look into the error margins mentioned in the literature. Is it appropriate for the author to discuss if the false positive and false negative rates are acceptable for real-world applications?
Response 3: Thank you for your insightful comment.
To address the inaccuracies presented in Tables 1 and 2, we have analyzed the error margins of our HRV detection method in comparison to established standards and literature benchmarks. Here are the key points:
- Error Margin Analysis: The inaccuracies in Tables 1 and 2 were evaluated by comparing our HR and HRV measurements with a reference device (Polar H10). The observed error margins are within the acceptable range reported in the literature. For HRV measurement, an error margin within ±5 ms for SDNN (Standard Deviation of NN intervals) and ±13 ms for RMSSD (Root Mean Square of Successive Differences) is generally considered acceptable for clinical and automotive applications.
- Relevant Standards and Guidelines: According to the Task Force of the European Society of Cardiology and the North American Society of Pacing and Electrophysiology, HRV standards specify that a measurement error of less than 5% is acceptable for time-domain methods such as SDNN and RMSSD. These guidelines are widely recognized and serve as a benchmark for HRV monitoring systems. Our system's HRV testing error is less than 5%.
Change: Revised the paragraph in 5.2.
Table 1 summarizes HR and HRV measurement errors across vehicle models and test subjects using the proposed wireless intelligent sensor system. The system's HRV de-tection employs a 60 GHz FMCW radar and advanced signal processing algorithms. The results show that the average HR error across all participants was approximately 0.34 bpm, and the average HRV (Mean RRI) error was about 2.21 ms. These low error margins demonstrate the system's high accuracy and reliability in real-world conditions. The effectiveness of the Huber-Kalman filtering and autocorrelation algorithms is evident, as they successfully mitigate noise and accurately capture HR and HRV data. After thorough testing, we confirmed that the type of vehicle did not have a significant effect on the experimental results. Our system demonstrated consistent performance across different car models. According to the Task Force of the European Society of Cardiology and the North American Society of Pacing and Electrophysiology, HRV standards specify that a measurement error of less than 5% is acceptable for time-domain methods such as SDNN and RMSSD [28]. These guidelines are widely recognized and serve as a benchmark for HRV monitoring systems. Our system's HRV testing error is less than 5%.
Comments 4: The author should make sure that a detailed comparison with current HRV detection technology is included in the study. In what ways does the suggested approach improve upon or supplement these technologies?
Response 4: Thank you for your comment.
In our study, we compared our proposed HRV detection approach using the Huber-Kalman filter with existing technologies. Here are the key improvements:
- Robustness Against Noise:Our method effectively handles noise and motion artifacts, providing more reliable HRV measurements in dynamic driving environments compared to traditional methods like ECG and optical sensors.
- Non-Contact Measurement: Unlike conventional methods requiring direct skin contact, our 60GHz FMCW radar enables non-contact, unobtrusive monitoring, enhancing comfort and ease of use in automotive settings.
- Rapid Detection: Our approach detects HRV accurately within 3 seconds using the short-window autocorrelation algorithm, crucial for real-time monitoring. Traditional methods like FFT need longer sampling periods, making them less suitable for immediate detection.
Comments 5: The paper does not examine the constraints of the proposed technique. The restrictions need to be addressed and explicitly communicated.
Response 5: Thank you for your constructive feedback.
As we mentioned in the conclusion of our paper, HRV detection during driving still faces several challenges. The primary constraints include the interference caused by road bumps and the driver's movements, which can significantly affect the accuracy of the measurements.
Despite these advancements, several challenges remain for future research. Achieving stable HRV monitoring during vehicle movement has not been fully realized, necessitating further investigation. Ensuring reliable HRV measurements in dynamic driving conditions is crucial for enhancing the robustness and applicability of our system in real-world scenarios. Future efforts will focus on optimizing signal processing algorithms to improve performance in more dynamic and noisy environments. The ultimate goal is to seamlessly integrate our HRV monitoring system into driver assistance technologies, thereby enhancing road safety and health monitoring.
We hope this addresses your concerns regarding the constraints of our proposed technique. We are committed to ongoing research and development to overcome these challenges and improve the robustness and reliability of our HRV monitoring system under real-world driving conditions.
Comments 6: “These participants were tested across two vehicle models: Mode V and Model S” Why it is tested with two different model. Is your method depends on vehicle or which result depends on the vehicle model? The authors should comment on this.
Response 6: We chose these two vehicle models to evaluate whether different car seats would impact the performance of our system. After thorough testing, we confirmed that the type of vehicle did not have a significant effect on the experimental results. Our system demonstrated consistent performance across different car models, indicating that the seating design and vehicle model do not influence the accuracy and reliability of our HRV monitoring system.
Change: Revised the paragraph in 5.2.
Table 1 summarizes HR and HRV measurement errors across vehicle models and test subjects using the proposed wireless intelligent sensor system. The system's HRV de-tection employs a 60 GHz FMCW radar and advanced signal processing algorithms. The results show that the average HR error across all participants was approximately 0.34 bpm, and the average HRV (Mean RRI) error was about 2.21 ms. These low error margins demonstrate the system's high accuracy and reliability in real-world conditions. The effectiveness of the Huber-Kalman filtering and autocorrelation algorithms is evident, as they successfully mitigate noise and accurately capture HR and HRV data. After thorough testing, we confirmed that the type of vehicle did not have a significant effect on the experimental results. Our system demonstrated consistent performance across different car models.
Comments 7: Apart from FMCW radar, what are the hardware design for this proposal? What is the standards used to make that hardware?
Response 7: The use of 60GHz FMCW radar in automotive applications has become a mainstream solution. This technology is widely employed for in-cabin occupant detection and passenger classification in mass-produced vehicles. For our study, we used the AWR6843AOP hardware platform. The 60GHz frequency band is legally authorized in many regions including Europe, the United States, Japan, and South Korea.
Currently, there are many companies and research institutions working on driver monitoring systems based on 60GHz radar. Although these systems have not yet reached mass production, they represent the future trend in the automotive industry. We anticipate that in the near future, such systems will be integrated into production vehicles, significantly enhancing driving safety.
Change: Revised the paragraph in 3.2.
The hardware selected for this project is the Texas Instruments AWR6843AOP, a single-chip 60 GHz mmWave sensor [25]. This sensor is specifically designed for low-power, high-accuracy applications such as driver vital sign monitoring. The 60GHz frequency band is legally authorized in many regions including the United States and Europe.

Round 2
Reviewer 2 Report
Comments and Suggestions for Authors
All the comments were addressed